# Boosted output performance of triboelectric nanogenerator via electric double layer effect

Jinsung Chun[1], Byeong Uk Ye[1], Jae Won Lee[1], Dukhyun Choi[2], Chong-Yun Kang[3,4], Sang-Woo Kim[5], Zhong Lin Wang[6] & Jeong Min Baik[1]

For existing triboelectric nanogenerators (TENGs), it is important to explore unique methods to further enhance the output power under realistic environments to speed up their commercialization. We report here a practical TENG composed of three layers, in which the key layer, an electric double layer, is inserted between a top layer, made of Al/poly-dimethylsiloxane, and a bottom layer, made of Al. The efficient charge separation in the middle layer, based on Volta's electrophorus, results from sequential contact configuration of the TENG and direct electrical connection of the middle layer to the earth. A sustainable and enhanced output performance of 1.22 mA and 46.8 mW cm$^{-2}$ under low frequency of 3 Hz is produced, giving over 16-fold enhancement in output power and corresponding to energy conversion efficiency of 22.4%. Finally, a portable power-supplying system, which provides enough d.c. power for charging a smart watch or phone battery, is also developed.

[1] School of Materials Science and Engineering, KIST-UNIST-Ulsan Center for Convergent Materials, Ulsan National Institute of Science and Technology (UNIST), Ulsan 689-798, Korea. [2] Department of Mechanical Engineering, College of Engineering, Kyung Hee University, Seocheon-dong, Giheung-gu, Yongin-si 446-701, Korea. [3] KU-KIST Graduate School of Converging Science and Technology, Korea University, Seoul 02841, Korea. [4] Center for Electronic Materials, Korea Institute of Science and Technology (KIST), Seoul 02792, Korea. [5] School of Advanced Materials Science and Engineering, Sungkyunkwan University (SKKU), Suwon 440-746, Korea. [6] School of Materials Science and Engineering, Georgia Institute of Technology, Atlanta, Georgia 30332-0245, USA. Correspondence and requests for materials should be addressed to J.M.B. (email: jbaik@unist.ac.kr).

nergy harvesting technologies enabled by the contact electrification that occurs when two objects are brought in contact and then separated, have been investigated as a means to efficiently convert mechanical energy into electricity[1–5]. Lightning may be one representative energy source initiated by friction. Within a thundercloud, many small ice particles collide with each other as they move around, generating electrical charges. Generally, positively charged particles in the cloud move up and negatively charged particles move down, determined by their weight; thus, the charges are separated. When the charge separation is produced enough, lightning occurs between the two charges within the cloud or between the cloud and the ground. Because the lightning carries a huge amount of energy of several billion joules, there have been several attempts to harvest the lightning energy for electricity[6,7]. However, very large-scale constructions are required to harvest it, and it is also hard to obtain high energy conversion efficiency because of the extremely high voltage generated.

Recently, a new type of power generating device, termed the triboelectric nanogenerator (TENG), based on triboelectric effects coupled with electrostatic effects, has been demonstrated[8–17]. So far, the TENGs have been already demonstrated in many applications such as self-powered chemical sensors[18–20], self-powered electrochemical processes[21–23] and powered commercial light-emitting diodes (LEDs)[24–27]. If the converted energy is convenient enough to power a number of small electronic devices such as smart phones/watches and tablets, we may use them without a battery in sight or the battery may not need to be replaced. Generally, the TENG in vertical-contact mode consists of two materials, chosen according to the difference in surface potentials, a metal and a dielectric in general[28–31]. During the friction between the two surfaces, the negative charges (that is, electrons) are transferred to the dielectric, inducing the flow of electrons of equal number through the external circuit. Thus, the charge density on the surface of the dielectric, which is constant if there is no charge loss by the air, determines the electric potentials, thereby, the output power. So far, the roughening of both surfaces has frequently been employed to increase the effective contact area and has been quite effective. Recently, there have been a few attempts to modify the properties of the dielectrics, such as the increase of the compressibility[32] and surface potential control[33–35]. Artificial injection of ions, such as corona discharging, was also considered as one means to maximize the charge density[36]. However, any dielectric has a maximum charge density that can be sustained on the surface and the capability seems to be not easily improved. Also, it may include limtations for long-term stability[36]. Thus, the detailed understanding of the working principle, and further progress in device technologies and triboelectric materials, is necessary to further enhance the performance.

Here we demonstrate a new type of TENG that overcomes the limit of the output performance of the conventional TENG under realistic environments. This work started with a basic idea to find how the two opposite charges are efficiently separated on the top and bottom electrodes during friction, as observed in thunderclouds. The conventional TENG has only positive charges on the electrodes, limiting the output power. The key component is a middle layer with Al film coated by Au nanoparticles, inserted between a top layer with mesoporous dielectric film on the top electrode (Al), and a bottom layer (Al). Thus, the new TENG is composed of three layers. As the force is applied to the top layer and then withdrawn, the positive and negative charges are spatially separated in the middle layer by sequential contact configuration of the TENG and direct electrical connection of the middle layer to the earth (ground), forming an electric double layer. This induces the positive and negative charges on the top and bottom electrodes, respectively, enhancing the electric potential. As the force is then withdrawn, the three layers are simultaneously separated, and more electrons are flowed through the external circuit by the enhanced potential. A sustainable and enhanced output performance of 1.22 mA and 46.8 mW cm$^{-2}$ under a low frequency of 3 Hz and compressive force of 50 N is produced, giving 16-fold enhancement in output power and corresponding to energy conversion efficiency of ∼22.4%. Wireless sensing systems such as the remote controller and the infrared sensor are successfully demonstrated with a signal-processing circuit. We also develop a portable power-supplying system that provides enough continuous d.c. power to charge a battery in smart watch/phone.

## Results

**Fabrication of TENG and the electrical outputs.** The schematic diagrams of the three-layer structured TENG are shown in Fig. 1a and detailed information is described in the Methods. The TENG consists of three layers: a top layer with mesoporous polymer film on the top electrode (Al), a middle layer with Al film coated by Au nanoparticles and a bottom layer (Al). Figure 1b shows that the mesoporous polydimethyl siloxane (PDMS) film, having a pore size of 1 μm, is uniformly formed on the top electrode. Previously, we showed that the mesoporous film is so effective in generating high output power because it exhibits more compressibility than the flat film[32]. The Au nanoparticles, having an average size of 100 nm, are uniformly coated on the Al film, as shown in Fig. 1c, which is used as a positive triboelectric material. Actually, the Au nanoparticles increase the effective contact area with the polymer layer and enhance the stability because of their high oxidation resistance. To maintain a gap between the top and middle layers, four springs were used, anchored at the edges. Finally, another Al film was used as a bottom electrode and four springs were also used to maintain a gap between the middle and bottom layers.

For achieving sequential contact configuration between the three layers, four springs with larger spring constants were used between the middle layer and bottom layer. Supplementary Figure 1 shows the optical images of the TENG when it is pressed and then released. When a compressive force is applied on the top layer, the mesoporous film is first brought into contact with the middle layer. When the force is continuously applied, the two layers make contact with the ground tip and then the bottom layer sequentially. As the force is withdrawn, it is clearly observed that the three layers are simultaneously separated. According to the coupled spring model[37], if there are no damping forces present, the upward restoring force is simultaneously exerted on the top layer and the middle layer. Finally, the middle layer is then detached from the ground tip. To compare the output performance, and confirm the advantage of the new TENG over a conventional TENG, a separate conventional TENG without the middle layer was also fabricated.

The output voltage and current measurements of the three-layer structured TENG were carried out under a cycled compressive force of 50 N and at an applied frequency of 3 Hz, plotted in Fig. 2a,b. The TENG has an active area of around 2 × 2 cm$^2$. We think that the working condition gives an input energy to the TENG lower than those in most previously reported TENGs[16]. It is well known that the output power increases with the magnitude of the force and frequency, and eventually saturates[16]. As a reference, we fabricated a two-layer structured TENG and the electrical signals were measured under the same conditions, as plotted in Supplementary Fig. 2. With a gap distance ($d_{gap}$) of 0.5 cm between the mesoporous film and Al

electrode, the TENG produced an alternating-current (a.c.) output with a short-circuit current ($I_{sc}$) of 0.095 mA and an open-circuit voltage ($V_{oc}$) of 100 V. As the gap size increased, the electrical signals increased up to 0.12 mA and 120 V at a gap distance of 1.5 cm. It is ascribed to the increase of the surface charge density on triboelectric materials due to the larger electric potential between the two materials; that is, the maximum charge density is obtained at the maximum of $d_{gap}$ (ref. 38). In the three-layer structured TENG with a gap distance of 1.5 cm between the top and the bottom electrodes, it is clearly observed that the output voltage and current reaches a record value of 300 V and 1.22 mA under the same compressive force, giving over 16-fold enhancement, compared with the two-layer structured TENG of same gap size. The Au decoration increases both output voltage and output current, although the enhancement is not signficant, compared with those in the three-layer structured TENG, shown in Supplementary Fig. 3. As a dielectric, we also used polytetrafluoroethylene film and the enhancement of the output power was also observed, meaning that this phenomenon occurs in general, shown in Supplementary Fig. 4.

**Power generation mechanism.** The marked enhancement in the output power of three-layer structured TENG was found to originate from the connection of the middle layer to the ground. We measured the output signals of the TENG without the ground connection, in which the middle layer made direct contact with the bottom layer, not the ground. It is clearly seen that the TENG produces an $I_{sc}$ of 0.15 mA and $V_{oc}$ of 80 V, which are the same as, or smaller than, those of conventional TENGs.

Supplementary Movie 1 also clearly shows that enhancement of the electrical signal by the ground connection under a high frequency of 10 Hz and compressive force of 50 N is produced. We also measured the electroluminescence (EL) of the single green LED powered by the TENG, the two-layer structured TENG and the three-layer structured TENGs, with and without the ground connection, as plotted in Fig. 2d. Without the ground connection, the EL intensity in the three-layer structured TENG is lower than that of the two-layer structured TENG. As the ground is connected, the EL intensity increases up to 2.56 times. However, the enhancement of the EL intensity is not as large as expected from the output current ($\sim$10 times). Actually, the EL intensity is not continuous, due to the generation of the instantaneous ($\sim$3 ms) output power of the TENG. Here a spectrometer (ELT-1000) with an integration time of 10 ms was used to obtain reliable EL signals. This equipment measures the signal every 1 s, which makes it possible to measure the instantaneous EL signal from the LED. The green EL images in the inset of Fig. 2d, and the red and blue EL images in Supplementary Movies 2 and 3 clearly show that the LEDs powered by the TENG with the ground connection are much brighter.

In general, the power generation of the TENG in vertical-contact mode by the compressive force can be understood from the coupling of the triboelectric effect and electrostatic induction, in which the electrons flow back and forth between the electrodes in a.c. characteristics through the external circuit, as shown in Fig. 2a,b. The a.c. signal observed in the three-layer structured TENG may indicate that the power generation mechanism is similar to that of the two-layer structured TENG. The working mechanism can be estimated from the physical movement of each

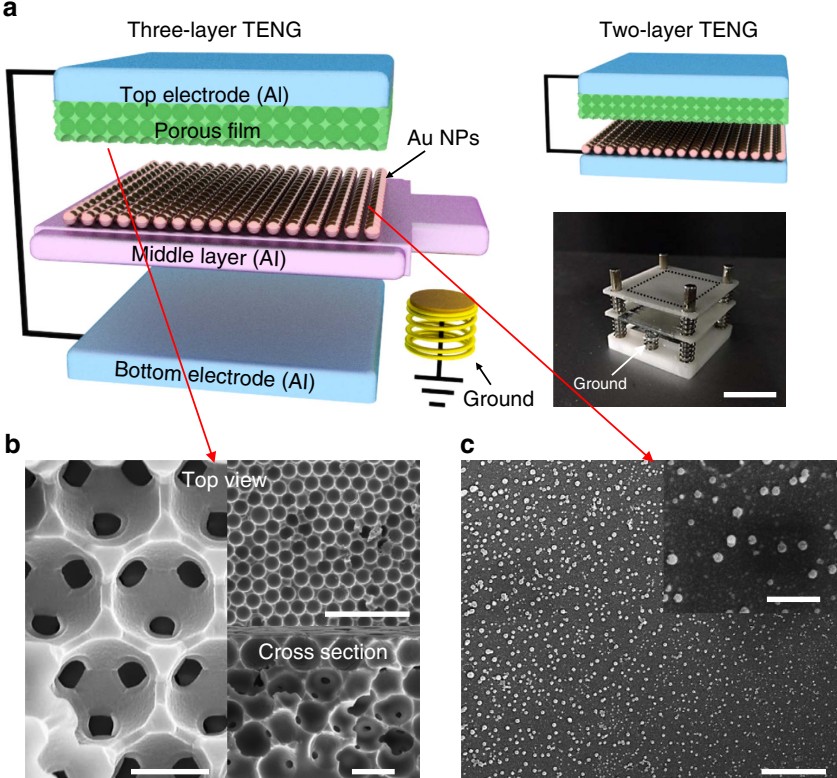

**Figure 1 | Fabrication of three-layer structured triboelectric nanogenerator.** (**a**) Schematic diagrams of the three-layer structured and two-layer structured triboelectric nanogenerator (TENG). The photograph of the nanogenerator is also shown. Scale bar, 1 cm. (**b**) Scanning electron microscope (SEM) images of the mesoporous polymer film on the top electrode. Top-view SEM images in left side and top-right corner, and cross-sectional view SEM image in bottom-right corner with scale bars of 1, 50 and 10 µm, respectively. (**c**) Top-view SEM images of the middle layer with Al film coated by Au nanoparticles (Au NPs). Scale bar of 5 µm. The inset also shows the expanded view, with scale bar of 1 µm.

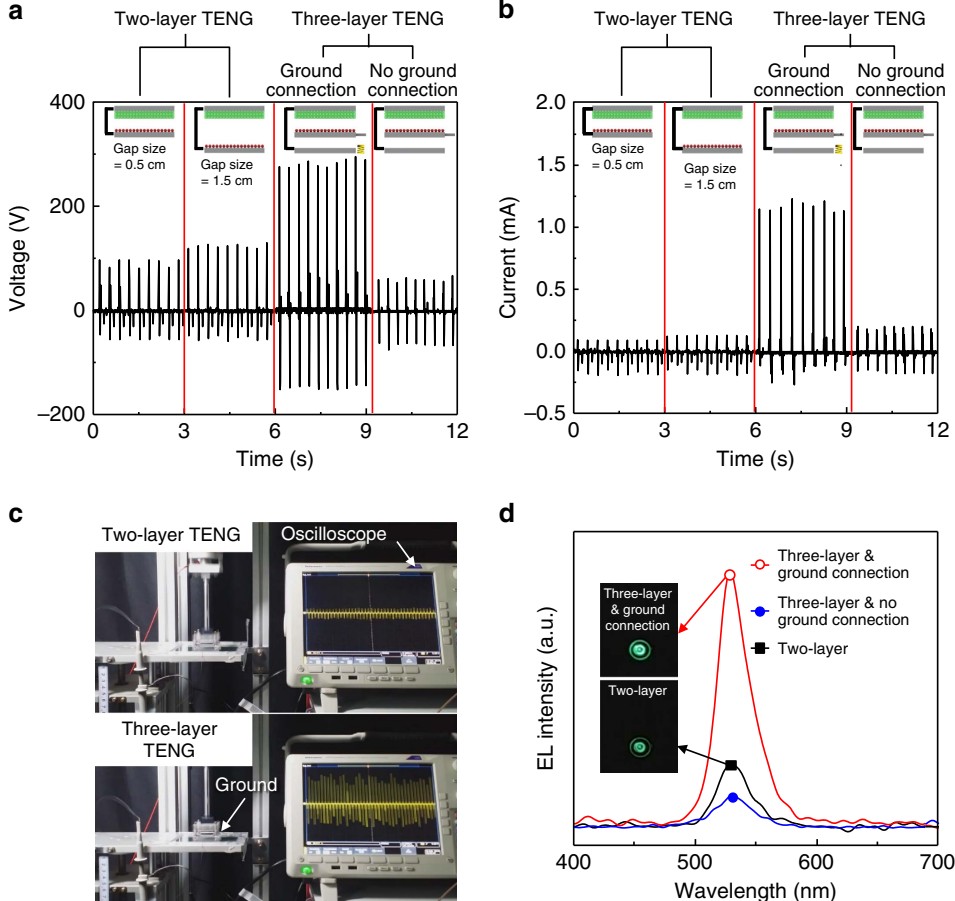

**Figure 2 | Electrical outputs of triboelectric nanogenerator. (a)** Output voltages and (**b**) currents of two-layer structured TENGs with gap sizes of 0.5 and 1.5 cm, and three-layer structured TENGs with and without a ground connection. (**c**) Optical images of measuring output signals for three-layer structured TENGs with and without a ground connection. (**d**) Electroluminescence (EL) spectra for commercial green LEDs powered by two-layer and three-layer structured TENGs as a function of wavelength.

layer when they are pressed and then released (Supplementary Fig. 1). When an external force is applied on the top layer, the porous film and the middle layer are brought into contact, resulting in positive charges on the surface of the middle layer and negative charges on the porous film of the top layer. When the force is continuously applied, the two layers make contact with the ground tip and then the bottom layer sequentially. As the force is withdrawn, the three layers are simultaneously separated. The positive charges in the middle layer will induce the flow of the electrons from the ground and the negative charges of the porous film will induce the positive charges on the top electrode, resulting in electron flow through the external circuit. After the first cycle, the top electrode is positively charged with the porous film negatively charged, while the bottom electrode is negatively charged, as shown in Supplementary Fig. 5.

In Fig. 3, after the first cycle, when an external force is applied on the top layer again, the negative charged porous film electrostatically induces positive charges on the surface of the middle layer, while negative charges are induced on the opposite surface of the layer, well known in Volta's electrophorus, known as the first electrostatic generator[39]. When the force is continuously applied, the negative charges of the middle layer induce positive charges on the bottom layer, leading to the electron flow through the external circuit. And then, the middle layer makes contact with the ground tip and the electrons go to the ground, evident by a current through the circuit connected to the ground, as shown in the inset. As the force is withdrawn, the

three layers are simultaneously separated and the electrons enter into the middle layer from the ground. The flow of electrons in the opposite direction is clearly shown in the inset. This is totally different from the charge generation occuring in a conventional TENG and very similar to the charge separation inside the thundercloud, as shown in Fig. 3b. Thus, the power generation mechanism can explain substantially larger electric potential, ideally twice, of the new TENG, compared with a conventional TENG.

To support the above-proposed working mechanism, the COMSOL simulations are performed for the two-layer and three-layer structured TENGs (Supplementary Fig. 6). The material parameters of the Al and PDMS, taken from the COMSOL simulation software, are used for the finite element analysis. The dielectric constants of Al and PDMS are 1 and 2.4, respectively. When the TENG is fully released, if we assume the electric potential ($U_{bottom}$) of the surfaces of the bottom layer to be zero, the electric potential of the surfaces of the top dielectric layer ($U_{top}$) can be expressed by $U_{top} = \sigma d_{gap}/\varepsilon_0$, where $\sigma$ is the triboelectric charge density, $\varepsilon_0$ is the vacuum permittivity of free space ($8.854 \times 10^{-12}$ F m$^{-1}$). For two-layer structured TENG, the bottom layer is assumed to be in neutral state as the TENG is fully released, that is, $U_{bottom}$ is 0. The maximum voltage drop can be defined by $\Delta V_{two} = V_{bottom} - V_{top} = 0 - V_{top} = -V_{top} = \sigma d/\varepsilon$. In the case of three-layer structured TENG, positive charges are induced in bottom electrode, thus, the potential is not zero. This leads to larger electric potential difference ($\Delta V_{three} = 2\Delta V_{two} = 2\sigma d/\varepsilon$)

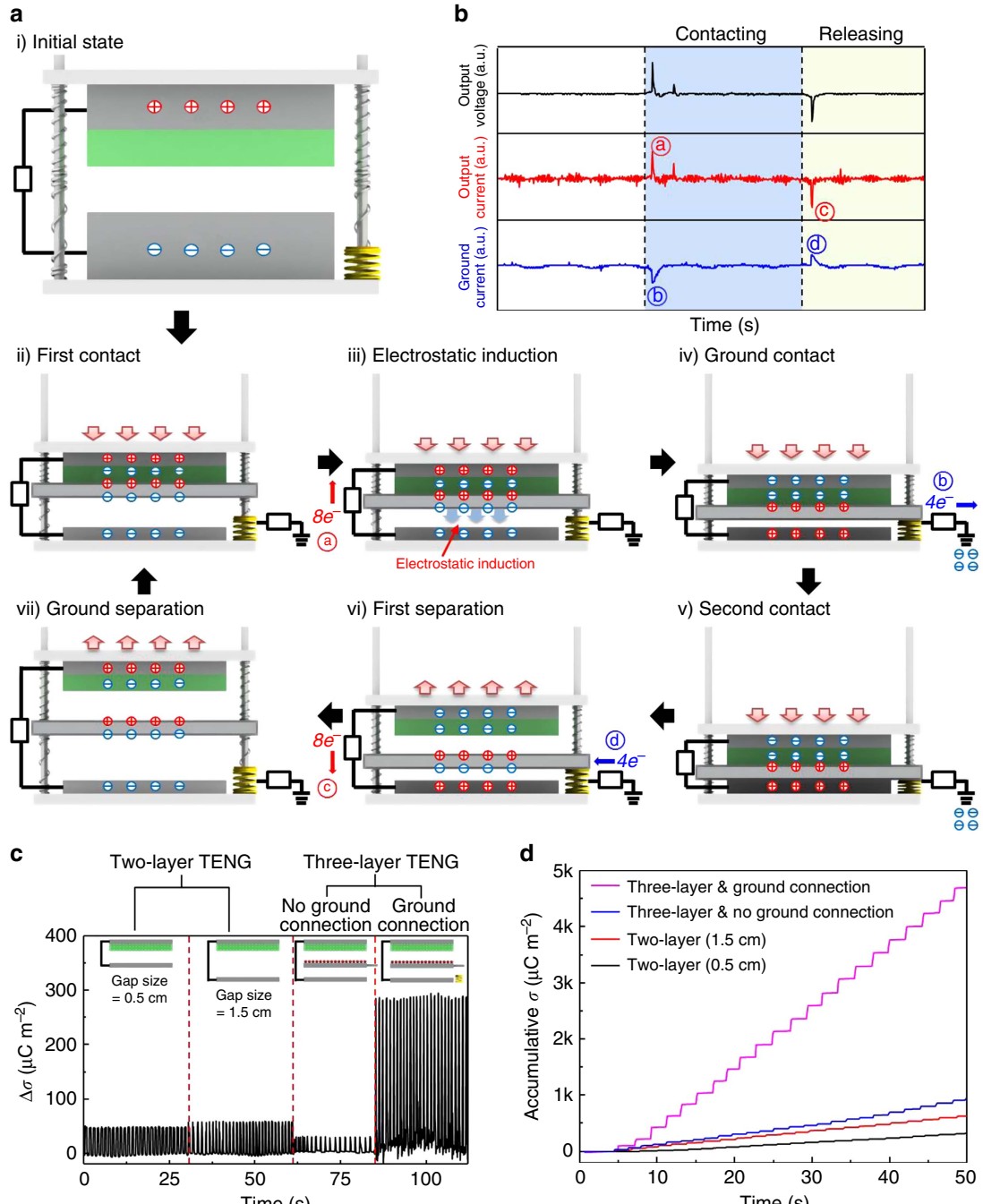

**Figure 3 | Working mechanism of three-layer structured triboelectric nanogenerator.** (**a**) Working mechanism for the generation of output voltage and current in the TENG under external force. (**b**) The output voltage and current produced by the TENG, and the current measured between the middle layer and ground. (**c**) The charge densities and (**d**) the accumulative charge densities at two-layer structured TENGs with gap sizes of 0.5 and 1.5 cm, and three-layer structured TENGs with and without a ground connection.

between the top and bottom layer, in good agreement with the experimental results of Fig. 2.

However, the instantaneous output current was increased by more than 10 times, not twice, compared with two-layer structured TENG. To see why the output current is correct, we measured the charge density on the surface of both electrodes in the three-layer structured TENG by using the electrometer system (Keithley 6514), $\sim 270\,\mu\mathrm{C\,cm^{-2}}$ under an external force of 50 N, as shown in Fig. 3d. The maximum surface charge density ($\sigma_{\mathrm{max}}$) can be also obtained from the theoretical analysis by comparing the threshold voltage for the air breakdown and voltage drop

across the airgap in the TENG. The $\sigma_{\mathrm{max}}$ can be expressed as below[40]

$$\sigma_{\mathrm{max}} = \left( \frac{A P \varepsilon_0 (d + x \varepsilon_{\mathrm{r}})}{d(\ln(Px + B))} \right) \min \qquad (1)$$

where $A$ and $B$ are the constants determined by the composition and pressure of the gas ($A = 2.87 \times 10^5\,\mathrm{V\,(atm\,m)^{-1}}$ and $B = 12.6$) and $P$, $d$, $x$, $\varepsilon_0$ and $\varepsilon_{\mathrm{r}}$ the pressure of the gas (101 kPa), the dielectric thickness, the airgap distance, the permittivity of free space, and the dielectric constant of dielectric, respectively. From the equation (1) and parameters of

this experimentally studied case, the theoretical $\sigma_{max}$ is numerically calculated to be 275.44 $\mu C\,m^{-2}$, in agreement with the experimentally measured value in Fig. 3c. The value is $\sim 4.5$ times larger than 60 $\mu C\,m^{-2}$ of two-layer structured TENG. Although the enhancement in charge density is not as big as the instantaneous output current, it seems that more electrons are flowed through the external circuit by the middle layer. The extraordinary increase in output current was also reported in a previous paper, but, it was not well understood[41]. Although additional studies are required, we ascribe the increase to the enhancement of the electric field between the top and bottom layers, due to the generation of the two opposite charges. Actually, the electrostatic force is proportional to the electric field strength[42]. Through a diode bridge to rectify the alternating output signals, the accumulative charge density (accumulative $\sigma$) of the three-layer structured TENG stably reached up to 4,600 $\mu C\,m^{-2}$ in 50 s (Fig. 3e).

**Energy conversion efficiency and output power of the triboelectric nanogenerator.** Figure 4a shows the output voltage and the output current of the three-layer structured TENG at a load resistance of 10 M$\Omega$, as a function of the pushing force from 10 to 90 N. Under a compressive force of 10 N, the TENG produces an $I_{sc}$ of 0.87 mA and $V_{oc}$ of 185 V. As the force increases to 90 N, it is clearly seen that the output voltages and currents increase up to 355 V and 1.5 mA. This is ascribed to the increase in the surface contact area, resulting in a larger surface charge density, which is a well-known effect[32,40]. Also note that the output voltages and currents are almost saturated at $\sim 60$ N, which may imply that the force is enough for maximizing the triboelectric charges on the surface of the mesoporous film.

Figure 4b shows the stability and durability test of the TENG under cycled compressive force of 30 N. It is clearly seen that the output current does not appear to change significantly after 10,800 cycles (60 min) although there is a small change in the output current. This result reveals the robustness and mechanical durability for a practical nanogenerator. Figure 4c shows the relationships between the electrical output performances of the three-layer structured TENG with the triggering frequency from 1 to 10 Hz. At 1 Hz, the TENG produces an $I_{sc}$ of 0.55 mA and $V_{oc}$ of 148 V, which is higher than those of the two-layer structured TENG measured at an applied frequency of 3 Hz. The output signals increased up to 1.22 mA and 300 V. The active size was increased to $7 \times 7\,cm^2$ and the electrical outputs were measured, as plotted in Supplementary Fig. 7. The output current and voltage increased to 2.5 mA and 440 V, respectively. Although the electrical signals did not increase linearly with the area, it is clearly seen that the output power from the TENG is able to instantaneously light up 256 LEDs simultaneously.

The energy conversion efficiency (ECE, $\eta$) may be estimated from the electrical output signals. In general, the conversion efficiency is defined as the ratio between the output energy and the input energy, in which the output energy is produced by the TENG and the input energy is the energy applied to the TENG. Here, as an input energy, kinetic energy ($E_k$) can be defined with the velocity ($v$) and mass ($m$) of top layer, and calculated as $\frac{1}{2}mv^2$. The velocity ($v$) of the top layer with the frequency was monitored by using a video camera with 480 f.p.s. and the mass ($m$) was estimated as the mass (50 g) of the top layer, in which the mass of Al thin film and PDMS layer are negligible. Thus, the kinetic energy ranges from 0.18 to 18.48 mJ, increasing with the frequency. At low values of input kinetic energy, the output voltages and currents steeply increase with the kinetic energy. As

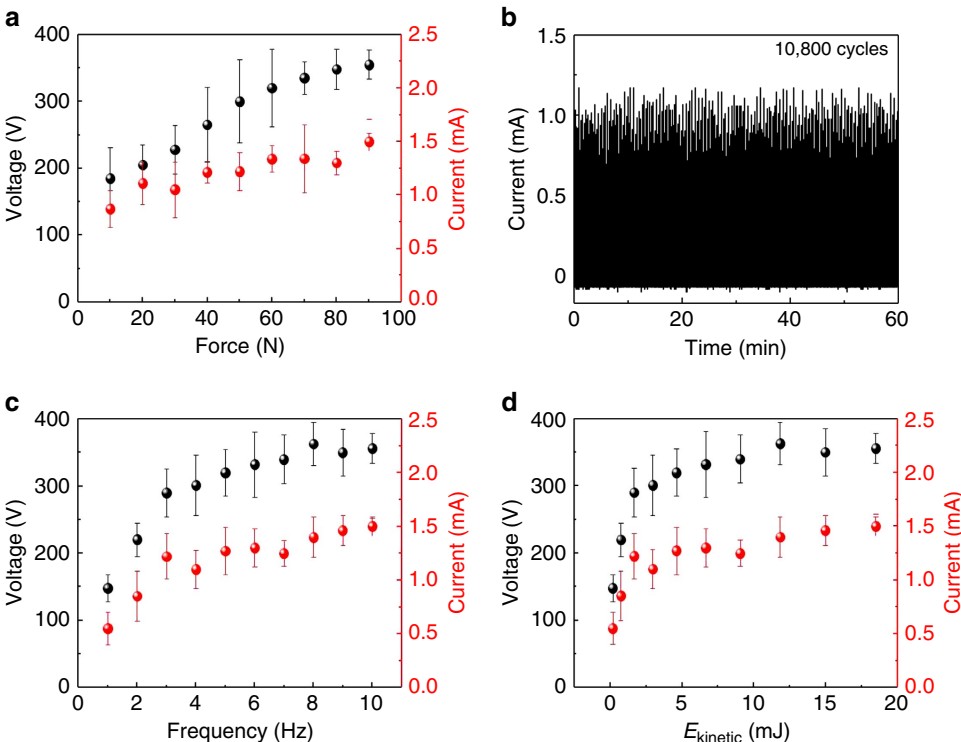

**Figure 4 | Electrical outputs of three-layer structured triboelectric nanogenerator under various forces and frequencies.** (**a**) The output current and voltage generated by three-layer structured TENGs under various forces from 10 to 90 N, (**b**) stability and durability test of the TENG under cycled compressive force of 30 N, (**c**) working frequencies from 1 to 10 Hz and (**d**) kinetic energies from 0.18 to 18.48 mJ. All error bars in the figure represent s.e.m. of the data.

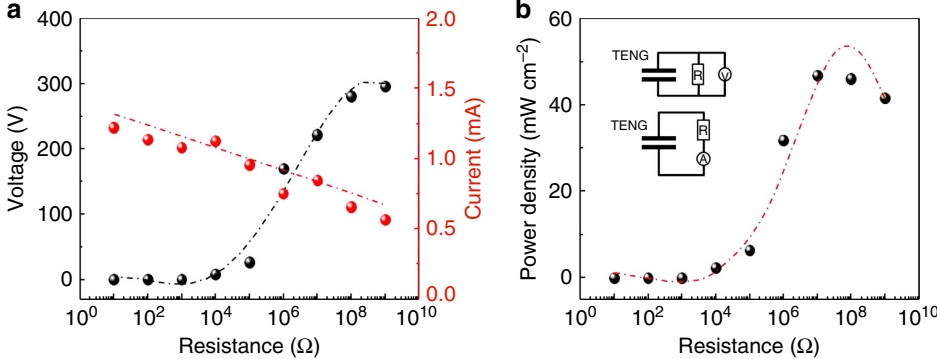

**Figure 5 | Output power of three-layer structured triboelectric nanogenerator and power density on the surface.** (**a**) The output voltage and current and (**b**) the output power of three-layer structured TENGs with the resistance of external loads from 10 to $10^9 \, \Omega$.

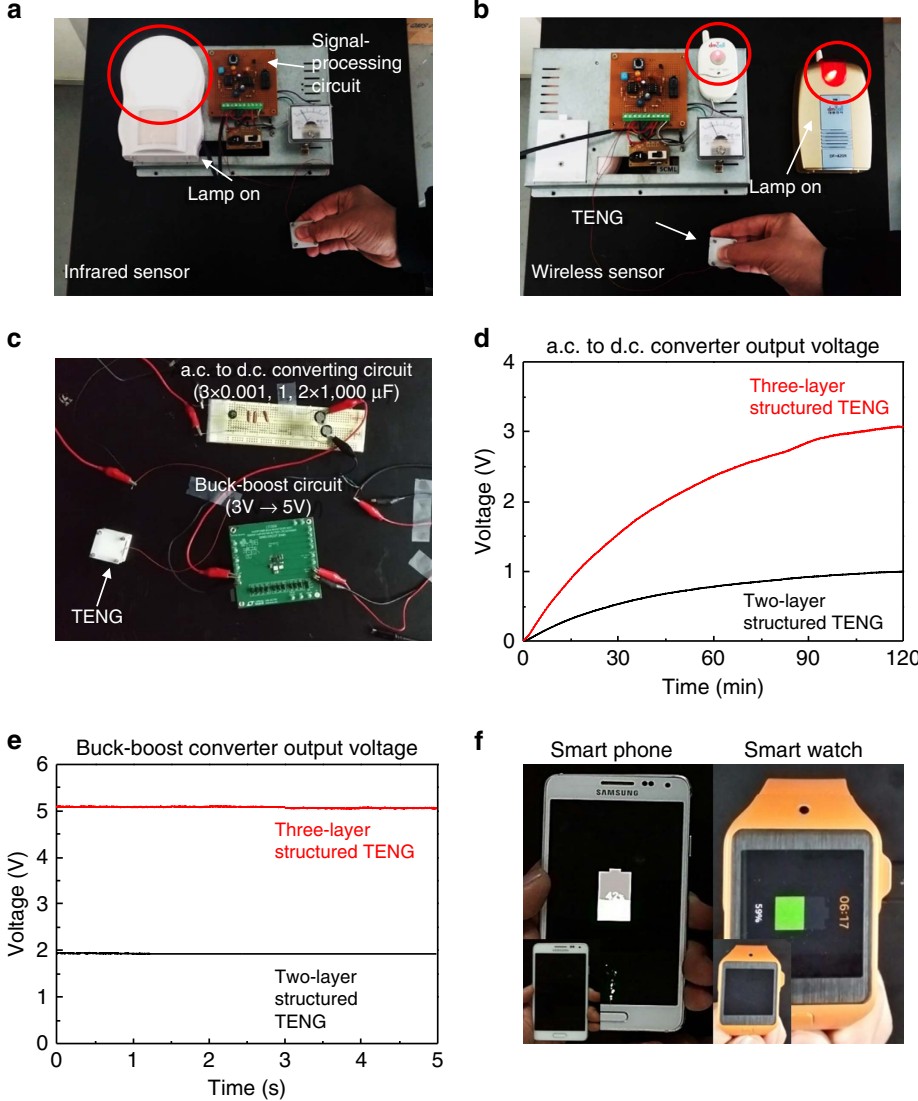

**Figure 6 | Wireless sensing system and portable power-supplying system for driving and charging electronics.** (**a**) Optical images of the infrared sensor and (**b**) the wireless sensor operated by the integrated wireless sensing system with a signal-processing circuit. (**c**) Optical images of the portable power-supplying system with an a.c. to d.c. converting circuit and a buck–boost circuit. (**d**) Measured output voltages in a a.c. to d.c. converter and (**e**) a buck–boost converter. (**f**) Optical images of charging a battery of smart phone/watch.

the input energy increases, the output signals increase very slowly, meaning that the efficiency can decrease at such a high input energy. Supplementary Fig. 8a and Supplementary Note 1 show

how to calculate the energy conversion efficiency when the compressive force of 50 N at a load resistance of 10 M$\Omega$ is applied to the top layer. Supplementary Figure 8b also shows the ECE

change of two- and three-layer structured TENGs as a function of applied kinetic energy from 0.18 to 18.48 mJ. In the three-layer structured TENG, as the input energy increases, the efficiency also increases up to 26% at 10 Hz, but there is no more significant increase in the efficiency over 5 mJ. It is clearly seen that three-layer structured TENG shows an ECE value 2–3 times higher than that of the two-layer structured TENG.

The output power of the three-layer structured TENG was also measured with external loads from $10\,\Omega$ to $1\,G\Omega$, as shown in Fig. 5a. The output voltage significantly increases with increasing the resistance, while the output current decreases. However, it is worthy of notice that the output current does not drop drastically with the resistance. Consequently, the instantaneous power is $\sim 46.8\,mW\,cm^{-2}$ at a resistance of $10\,M\Omega$, as shown in Fig. 5b, giving over 16-fold power enhancement, compared with the conventional TENG with same gap size, as shown in Supplementary Fig. 9a,b.

**Wireless sensing system and portable power-supplying system to smart watch or phone**. To demonstrate the capability of the three-layer structured TENG as a practical power source, a wireless sensing system was developed by integrating the TENG with a signal-processing circuit, as shown in Supplementary Fig. 10. The wireless sensing system can be easily operated by the output voltage generated from the TENG to trigger an integrated circuit timer (NE 555) that controls a wireless transmitter for remotely switching a siren between an emergency and a normal state. When the TENG is pushed by a human hand, the generated output voltage operates the remote controller and the infrared sensor, resulting in turning on a siren, along with the flashing light of the sensor, as shown in Fig. 6a,b, and Supplementary Movies 4 and 5.

Finally, a portable power-supplying system was also developed by integrating the TENG with an a.c. to d.c. converting circuit and buck–boost circuit, as shown in Fig. 6c and Supplementary Fig. 11. The converting circuit consists of three rectifiers and two capacitors ($3 \times 0.001$, 1, $2 \times 1,000\,\mu F$) which converts a.c. to d.c. output signal. When the compressive force of 50 N was applied to the TENG, the charged voltage of the capacitors was boosted up to a constant voltage of 5 V with using a buck–boost circuit, as shown in Fig. 6d,e. After 2 h, it is clearly seen that both smart phone and smart watch are being charged by connecting the batteries with the power-supplying system, as shown in Fig. 6f. It is also seen that the batteries are being charged in Supplementary Movies 6 and 7. Although it still takes a long time to be fully charged, it can provide a continuous uniform enough d.c. power to charge the battery and to drive various commercial electronics. However, under the same conditions, the two-layer structured TENG only produced a constant voltage of less than 2 V, not enough for charging the smart phone or smart watch.

## Discussion

In summary, we reported here a practical TENG composed of three layers, not two layers, by inserting a metallic middle layer between a top layer, made of Al/PDMS, and a bottom layer, made of Al. This TENG effectively conjoins two operation modes: vertical contact-separation mode and single-electrode mode. However, based on the power generation mechanism, the device design is different with the stack of TENGs with the two modes. A sustainable and enhanced output performance of 1.22 mA and $46.8\,mW\,cm^{-2}$ under a low frequency of 3 Hz and a compressive force of 50 N was produced, giving 16-fold enhancement in output power, compared with the conventional TENG, and corresponding to energy conversion efficiency of approximately

22.4%. As a key for enhancing the output power, we believe that the charge separation in the middle layer, known in Volta's electrophorus, is efficiently induced by sequential contact configuration of the TENG and direct electrical connection of the middle layer to the earth (ground), increasing the charge density on the top and bottom layers as the force is applied to the top layer. As the force is withdrawn, the three layers are simultaneously separated, and more electrons are flowed through the circuit by the enhanced potential.

Through the integration of the TENG with a signal-processing circuit, wireless sensors such as the remote controller and the infrared sensor were demonstrated. Finally, a portable power-supplying system was also successfully demonstrated by integrating the TENG with an a.c. to d.c. converting circuit and a buck–boost circuit, in which it provided enough continuous d.c. power to charge a battery of smart watch/phone. Although further improvement is needed, from these results, it is expected that the new TENG can contribute efficiently not only to the realization of self-powered electronics, but also possibly to the development of large power generation technologies on a large scale.

## Methods

**Production of dielectric layer and Au nanoparticle-coated Al layer**. PDMS (Sylgard 184, Dow Corning) was used as the polymer layer with many pores, which was prepared by the selective removal technique. The solution of mixed base monomer and curing agent in a mass ratio of 10:1 was dropped on the beaker. An aqueous suspension of polystyrene spheres (2.6 wt%, Polysciences, Warrington) was used to fabricate the PDMS inverse opal-structured film. Many layers of polystyrene spheres with diameters of $1\,\mu m$ were stacked in a face-centred cubic structure onto a $SiO_2$/Si substrate. Then, PDMS solution was poured into the periodically-arranged polystyrene spheres, and allowed to solidify into an amorphous free-standing film by heating it on a hotplate at $90\,°C$. Further, to obtain the PDMS inverse opal-structured film, the PDMS matrix was detached from the substrate, and soaked in acetone for 24 h to remove the polystyrene spheres. The effective area and thickness of both flat and porous structure films were $2 \times 2\,cm^2$ and $300\,\mu m$, respectively. As a dielectric, polytetrafluoroethylene film was also used to confirm that this design is broadly applicable to a range of dielectrics.

An aqueous suspension of 100-nm gold (Au) colloids (BBI International) was used for the fabrication of the Au nanoparticle-coated electrode. The Al electrode was first treated with Ultraviolet/Ozone (AHTECH LTS, South Korea) to make the surface of the substrate hydrophilic. Au colloid solution was spin coated onto the electrode's surface, and then the samples were dried for 12 h in a dry box at room temperature, followed by the attachment of an Al electrode on the opposite side of Kapton film, used as the middle layer.

**Fabrication of three-layer structured triboelectric nanogenerator**. The Al film was attached on an Acryl plate, used as the bottom layer. Four springs to support the middle layer and another Al-coated spring, which connected to the ground, were installed on the Acryl plate. The middle layer, which is a Au nanoparticle-coated Al film, was stacked on them. Four more springs were used to support the top layer. The top layer, which consists of a dielectric and Acryl plate, was then stacked on them.

**Characterization and measurements**. The morphologies of porous PDMS films and Au nanoparticle-coated electrode were further characterized by a field emission-scanning electron microscope. The light output power of LEDs was measured using an LED tester (ECOPIA Inc., model no. ELT-1000). A pushing tester (Labworks Inc., model no. ET-126-4) was used to create vertical compressive strain in the nanogenerator. A Tektronix DPO 3052 Digital Phosphor Oscilloscope and a low-noise current preamplifier (model no. SR570, Stanford Research Systems, Inc.) were used for electrical measurements. The charge density from the output signals was measured using a Keithley 6514 system electrometer. The energy harvesting power supply, consisting of an integrated low-loss full-wave bridge with a high voltage buck–boost converter circuit (Linear Technology Inc., model no. DC2048A) and an a.c. to d.c. converter (Texas Instruments Inc., trigger NE 555), harvested energy from three-layer structured TENG.

**Data availability**. All relevant data are available from the authors on request.

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

## Acknowledgements

This work was supported by Samsung Research Funding Center of Samsung Electronics under Project Number SRFC-TA1403-06.

## Author contributions

J.C. conceived the idea, analysed the data and wrote the main manuscript text. B.U.Y. and J.W.L. conducted output measurements and prepared figures. D.C., C.-Y.K., S.-W.K. and Z.L.W. provided advice for the research and revised the manuscript. J.M.B. conceived of and supervised this study, and provided intellectual and technical guidance. All authors discussed the results, wrote and commented on the manuscript.

## Additional information

**Competing financial interests**: The authors declare no competing financial interests.

**How to cite this article**: Chun, J. *et al.* Boosted output performance of triboelectric nanogenerator via electric double layer effect. *Nat. Commun.* **7,** 12985 doi: 10.1038/ncomms12985 (2016).

