## [Peer Review File · Nature Communications]

Reviewer #1 (Remarks to the Author):

Authors developed a tri-layered electrical-generating device through contact. The paper could be accepted after addressing the following issues.

. Since the main contribution of the work is to composite a three-layered structure, not two, as authors have repeatedly emphasized, there should be a comparative study of both. It is currently lacking. What is the efficiency of the 2-layered, 10%?

. Is the porous film monolayer? The SEM image in Fig. 1b seems to indicate so. Please clarify.

. The Fig. 3b looks like a cartoon. Authors are not serious about their scientific work.

.

Reviewer #2 (Remarks to the Author):

In this paper, the authors present a three-layer triboelectric nanogenerator (TENG) with high output performance. An electric layer made of Al foil covered by Au particles is placed as a middle layer between Al/mesoporous film and Al. As is claimed by authors, a 16-fold enhancement in output power was achieved due to the Volta's electrophorus. The fabricated device was also demonstrated for several practical applications. The research field of renewable power source and the research topic of this work is interesting. However, the multiple-layer TENG (Adv. Funct. Mater., 24, 4090, 2014), mesoporous PDMS film (Adv. Mater., 26, 5037, 2014; Energy. Environ. Sci., 8, 2015, 3006) and the electrode covered by Au particles (Angew. Chem. Int. Ed., 52, 5065, 2013), all of them have already been reported previously. Thus, the novelty of this work is not qualified to Nature Communications, and it is recommended to resubmit it to other journals.

Besides the above comments, the authors may consider the following comments before resubmission.

(I) The reason of using middle layer of Al foil covered by Au particles is claimed to be Volta's Electrophorus, but actually, the underlying mechanism is still based on triboelectrification and electrostatic induction. In other words, it is not a new phenomenon. Additionally, it is not convinced that the "middle layer" significantly enhances the output performance of the device. Furthermore, it is believed that the Au particles is the key point instead of the middle layer for this enhancement. Thus, in order to verify the effect of the middle layer, it does not make sense to compare the two-layer TENG with flat surface to the three-layer TENG with textured surface. In other words, the bottom Al electrode of the two-layer TENG should also be textured by coating Au particles. (II) The working principle of three-layer TENG is confusing. As is shown in Figure 3, both of negative and positive charges exist simultaneously on the surfaces of metal, is it true? For the metal, it should be only one kind of charges existing on the surface at the same time, right? (III) Some errors in Figures, such as the resistor connecting electrodes should be with the same size, etc.

Reviewer #3 (Remarks to the Author):

The manuscript described a new triboelectric nanogenerator whose performance was significantly enhanced by an additional layer inserted between two electrodes. This work is new and the presentation is clear. The conclusion was supported by the data, and reference is appropriate. This reviewer recommends the publication of this paper. The author might consider the following for minor revision.

1. Author mentioned that different springs were used so that contacts between 3 layers occurred in sequence. However, as the force was withdrawn, the author indicated that the separation occurred simultaneously (page 5, line 107-109). Why didn't the separation occur in sequence as well?
2. It is not clear how and why the author assume the areal charge density of the bottom electrode of two-layer device to be zero in the COMSOL simulation. Since the electrode is metallic, the surface should have non-zero areal charge density, such that the electric field within the metal is zero.
3. If we simply the device structure with parallel-plate model, the voltage difference between top and bottom electrode can be estimated as $\sigma \cdot d / \epsilon$. If the areal charge density of top electrode is σ on the top electrode, the voltage difference won't change with or without the third layer in the middle.
4. The video of charging phone and watch with TENG, TENG seems stationary. How can it generate electricity and charge phone and watch without being activated? If the phone and watch are charged from a capacitor in the video, the author might want to explain explicitly in the manuscript.

July 7, 2016

Dr. Ariane Vartanian
Assistant Editor, Nature Communications
The Macmillan Building
4 Crinan Street
London N1 9XW, UK

Re: Decision on manuscript NCOMMS-16-10644
Title: "Boosted Output Performance of Triboelectric Nanogenerator via Electric Double Layer Effect"
Authors: Jinsung Chun, Byeong Uk Ye, Jae Won Lee, Dukhyun Choi, Sang-Woo Kim, Zhong Lin Wang, and Jeong Min Baik*
Manuscript ID: NCOMMS-16-10644

Dear Dr. Vartanian,

We believe that the manuscript qualifies as a paper in Nature Communications. We think that we fully responded to the concerns and comments of the reviewers. Our reasons are as follows:

Reviewer #1

Reviewers' comments:

Authors developed a tri-layered electrical-generating device through contact. The paper could be accepted after addressing the following issues.

(1) Comment

Since the main contribution of the work is to composite a three-layered structure, not two, as authors have repeatedly emphasized, there should be a comparative study of both. It is currently lacking. What is the efficiency of the 2-layered, 10%?

Author reply: We strongly appreciate reviewer's valuable and helpful comments. The efficiency of the two-layer structured TENG was already included in Supplementary Figure 7 in manuscript to compare with that of three-layer structured TENG. (Fig. 1) Additionally, to help the understanding of the reviewers, Table 1 shows how to calculate the energy conversion efficiency (ECE, η) of two- and three-layer structured TENGs under frequency from 1 Hz to 10 Hz.

a i) Input energy (kinetic energy)

$$E_{elastic} = \frac{1}{2}mv^2 = 4.62 \text{ mJ}$$

ii) Output energy (electric energy)

$$E_{electric} = Q = \int_{t_1}^{t_2} I^2 R dt = 1.04 \text{ mJ}$$

iii) Energy conversion efficiency (η)

$$\eta = \frac{E_{electric}}{E_{kinetic}} \times 100\% = \frac{1.04 \text{ mJ}}{4.62 \text{ mJ}} \times 100\% = 22.4\%$$

Figure 1 | (a) Calculations in ECE (η) of two- and three-layer structured TENGs under frequency of 5 Hz. (b) The ECE (η) change of two- and three-layer structured TENGs as a function of applied kinetic energy from 0.18 to 18.48 mJ.

Frequency	Mass (g)	v (m/s)	$E_{kinetic}$ (mJ)	R (Ω)	$E_{electric}$ (mJ)		ECE (η)	
					Two layer	Three layer	Two layer	Three layer
1	50	0.086	0.185	10^7	0.004	0.018	2.082	9.701
2		0.172	0.739		0.019	0.111	2.527	14.992
3		0.258	1.663		0.071	0.358	4.282	21.518
4		0.344	2.957		0.141	0.574	4.758	19.402
5		0.430	4.620		0.270	1.035	5.840	22.400
6		0.516	6.653		0.396	1.525	5.948	22.929
7		0.602	9.055		0.637	1.996	7.032	22.047
8		0.688	11.827		0.934	2.920	7.899	24.693
9		0.774	14.969		1.127	3.855	7.532	25.751
10		0.860	18.480		1.502	4.889	8.129	26.457

Table 1 | Parameters for calculations of ECE (η) of two- and three-layer structured TENGs under frequency from 1 Hz to 10 Hz.

(2) Comment

Is the porous film monolayer? The SEM image in Fig. 1b seems to indicate so. Please clarify.

Author reply: We strongly appreciate reviewer's valuable and helpful comments. We are also sorry for the confusion. The porous film is not monolayer, 5 ~ 6 layers. To clarify it, we added 'Top view' in left side and top-right corner, and 'Cross-sectional view' in bottom-right corner in the Fig. 1b. We also modified the figure caption.

Figure 2 | Fabrication of three-layer structured triboelectric nanogenerator. (a) Schematic diagrams of the three-layer structured and two-layer structured triboelectric nanogenerator. The photograph of the nanogenerator is also shown. (b) SEM image of the mesoporous polymer film on the top electrode. Top-view SEM images in left side and top-right corner, and cross-sectional view SEM image in bottom-right corner. (c) Top-view SEM images of the middle layer with Al film coated by Au nanoparticles. The inset also shows the expanded view.

(3) Comment

The Fig. 3b looks like a cartoon. Authors are not serious about their scientific work.

Author reply: We strongly appreciate reviewer's valuable and helpful comments. According to the reviewer's comment, we removed the thundercloud image and the caption from the Fig. 3b to emphasize our scientific work.

Figure 3 | Working mechanism of three-layer structured triboelectric nanogenerator. (a) Working mechanism for the generation of output voltage and current in the nanogenerator under external force. (b) The output voltage and current produced by the nanogenerator, and the current measured between the middle layer and ground. (c) The charge densities and (d) the accumulative charge densities at two-layered structured triboelectric nanogenerators with gap sizes of 0.5 and 1.5 cm, and three-layered structured triboelectric nanogenerators with and without a ground connection.

Reviewer #2

Reviewers' comments:

(1) Comment

In this paper, the authors present a three-layer triboelectric nanogenerator (TENG) with high output performance. An electric layer made of Al foil covered by Au particles is placed as a middle layer between Al/mesoporous film and Al. As is claimed by authors, a 16-fold enhancement in output power was achieved due to the Volta's electrophorus. The fabricated device was also demonstrated for several practical applications. The research field of renewable power source and the research topic of this work is interesting. However, the multiple-layer TENG (Adv. Funct. Mater., 24, 4090, 2014), mesoporous PDMS film (Adv. Mater., 26, 5037, 2014; Energy. Environ. Sci., 8, 2015, 3006) and the electrode covered by Au particles (Angew. Chem. Int. Ed., 52, 5065, 2013), all of them have already been reported previously. Thus, the novelty of this work is not qualified to Nature Communications, and it is recommended to resubmit it to other journals. Besides the above comments, the authors may consider the following comments before resubmission.

Author reply: We strongly appreciate reviewer's valuable and helpful comments. As the reviewer commented, we showed a 16-fold enhancement in output power, compared with the two-layered TENG. It is really extraordinary enhancement that have not ever reported so far. Our goal here is to suggest a new TENG design, which ideally and intrinsically increase the output performance, for a world-record value. As the reviewer mentioned, we used two technologies, mesoporous film and Au nanoparticles, for the fabrication of the two-layered TENG which generates high output power. This is a reference device. For a world-record value, it may be natural to use all useful and valuable technologies reported so far. As the reviewer may know, the mesoporous film is a technology from my lab. Table 2 shows the summary of the papers mentioned by the reviewer¹⁻⁴.

However, please note that the multiple-layer TENG and our TENG is total different. Figures 2a and 2b in the manuscript will explain why they are totally different. Table 3 also shows the working mechanism of 3D stack integrated triboelectric nanogenerator (paper by the reviewer) and three-layer structured triboelectric nanogenerator (our TENG). We did not focus on the stacking, which was not unique. I believe that the reviewer will fully understand.

Reference	Structure	Materials	Outputs	Key idea	Comparisons
1	 3D stack	Negative: Nanowire-PTFE Positive: Al	303 V 1.14 mA 104.6 W/m ² (10.47mW/cm ²)	Output current increased by series-connected TENGs.	Ref.: Simple connection of TENGs to increase only output current. (i.e. Maximum output current is proportional to the number of TENGs) This work: Maximized internal voltage drop by inserting a middle layer (Significant enhancement in output current and voltage for a TENG.)
2	 Two-layer structured TENG	Negative: Porous PDMS Positive: Al	130 V 0.10 mA	High capacitance change by the increase in effective (ϵ/d) value for compressibility of porous structure.	This work: Key idea of Ref. 2 is applied to three-layer structured TENG. (To start at reference TENG structure with optimized and maximized efficiency)
3	 Two-layer structured TENG	Negative: Porous PDMS with Au nanoparticles Positive: Al	220 V 0.03 mA 13 mW (0.16 mW/cm ²)	The enhancement of the charge density and surface potential created by the contact between Au NPs and PDMS inside the pores.	This work: Key idea of Ref. 3 is applied to three-layer structured TENG. (To start at reference TENG structure with optimized and maximized efficiency)
4	 Two-layer structured TENG	Negative: PDMS Positive: Au nanoparticles decorated Au	105 V 0.06 mA (6.9 mW/cm ²)	i) Increased output with large contact area by Au nanoparticles decoration ii) Mercury ion detection because of the different triboelectric polarity of Au nanoparticles and mercury ions.	This work: Key idea of Ref. 4 is applied to three-layer structured TENG. (To start at reference TENG structure with optimized and maximized efficiency)

Structure	Mechanism
 3D stack	 Three-layer structured TENG	
(2) Comment

The reason of using middle layer of Al foil covered by Au particles is claimed to be Volta's Electrophorus, but actually, the underlying mechanism is still based on triboelectrification and electrostatic induction. In other words, it is not a new phenomenon. Additionally, it is not convinced that the "middle layer" significantly enhances the output performance of the device. Furthermore, it is believed that the Au particles are the key point instead of the middle layer for this enhancement. Thus, in order to verify the effect of the middle layer, it does not make sense to compare the two-layer TENG with flat surface to the three-layer TENG with textured surface. In other words, the bottom Al electrode of the two-layer TENG should also be textured by coating Au particles.

Author reply: We strongly appreciate reviewer's valuable and helpful comments. We found that the reviewer's comment was very appropriate. According to the reviewer's comments, we fabricated two-layer and three-layer structured TENGs with and without Au nanoparticles to verify the effect of Au nanoparticle decoration on the output power. As the reviewer commented, the Au decoration increases both output voltage and output current. However, it is clearly seen that by the Au decoration, we cannot have the significant enhancement in output power, comparable to that of the three-layer structured TENG. We changed all data of the two-layer TENG to those with the Au particles. We are also sorry for the confusion. According to the reviewer's comment, the bottom Al electrodes of all two-layer TENGs were also textured by coating Au particles of schematic images in Fig. 1, Fig. 2, and supplementary Fig. 3 in manuscript, as shown in Figs. 5, 6, and 7.

Figure 4 | The output voltages (a, b) and output currents (c, d) of two-layer and three-layer structured TENGs with and without Au nanoparticles. The measured charge densities are shown in (e) and (f).

Figure 5 | Fabrication of three-layer structured triboelectric nanogenerator. (a) Schematic diagrams of the three-layer structured and two-layer structured triboelectric nanogenerator. The photograph of the nanogenerator is also shown. (b) SEM image of the mesoporous polymer film on the top electrode. Top-view SEM images in left side and top-right corner, and cross-sectional view SEM image in bottom-right corner. (c) Top-view SEM images of the middle layer with Al film coated by Au nanoparticles. The inset also shows the expanded view.

Figure 6 | Electrical outputs of triboelectric nanogenerator. (a) Output voltages and **(b)** currents of two-layer structured triboelectric nanogenerators with gap sizes of 0.5 and 1.5 cm, and three-layer structured triboelectric nanogenerators with and without a ground connection. **(c)** Optical images of measuring output signals for three-layer structured triboelectric nanogenerators with and without a ground connection. **(d)** EL spectra for commercial green LEDs powered by two-layer and three-layer structured triboelectric nanogenerators as a function of wavelength.

Figure 7 | Output voltage in (a) and current in (b) of three-layer structured triboelectric nanogenerators with various with PDMS film and PTFE film.

(3) Comment

The working principle of three-layer TENG is confusing. As is shown in Figure 3, both of negative and positive charges exist simultaneously on the surfaces of metal, is it true? For the metal, it should be only one kind of charges existing on the surface at the same time, right?

Author reply: We strongly appreciate reviewer's valuable and helpful comments. Here, we believe that the Volta's electrophorus effect is a key idea to significantly enhance the output power. As the reviewer may know, the Volta's electrophorus consists of a polymer plate (dielectric) and a metal plate with an insulating handle. The principle is as follows; The polymer plate is charged by the triboelectric effect by rubbing and gains negative charges. The metal plate is then placed onto the polymer plate. The electrostatic field of the charged polymer causes the charges in the metal plate to separate. It develops two regions of charges, positive charges and negative charges. The positive charges in the plate are attracted to the side facing down toward the polymer, charging it positively, while the negative charges are repelled to the side facing up, charging it negatively, as shown in Movie 1 from the Youtube site (<https://www.youtube.com/watch?v=Hy-H4JOxETE>). The movie we uploaded will help the reviewer.

Movie 1 | The charges separated in the metal plate by the electrostatic field of the charged polymer.

(4) Comment

Some errors in Figures, such as the resistor connecting electrodes should be with the same size, etc.

Author reply: We strongly appreciate reviewer's valuable and helpful comments. We are also sorry for the confusion. According to the reviewer's comment, the resistors connected with the electrodes in the schematic images of Fig. 3, and supplementary Figs. 2 and 4 in manuscript were modified with same size. (Figs. 8, 9, and 10)

Figure 8 | Working mechanism of three-layer structured triboelectric nanogenerator. (a) Working mechanism for the generation of output voltage and current in the nanogenerator under external force. (b) The output voltage and current produced by the nanogenerator, and the current measured between the middle layer and ground. (c) The charge densities and (d) the accumulative charge densities at two-layered structured triboelectric nanogenerators with gap sizes of 0.5 and 1.5 cm, and three-layered structured triboelectric nanogenerators with and without a ground connection.

Figure 9 | Two-layer structured TENG and the electrical signals were measured under same condition.

Figure 10 | The working mechanism can be estimated from the physical movement of each layer when it is pressed and then released at 1st cycle.

Reviewer #3

Reviewers' comments:

The manuscript described a new triboelectric nanogenerator whose performance was significantly enhanced by an additional layer inserted between two electrodes. This work is new and the presentation is clear. The conclusion was supported by the data, and reference is appropriate. This reviewer recommends the publication of this paper. The author might consider the following for minor revision.

(1) Comment

Author mentioned that different springs were used so that contacts between 3 layers occurred in sequence. However, as the force was withdrawn, the author indicated that the separation occurred simultaneously (page 5, line 107-109). Why didn't the separation occur in sequence as well?

Author reply: We strongly appreciate reviewer's valuable and helpful comments. We also think that it is so important. At first, we also thought that the separation would occur in sequence, however, it could not explain the power generation mechanism suggested here. In Figure S1, we tried to show the separation occurred simultaneously although the photos were not clear. According to the coupled spring model, two-degree-of-freedom systems can be assumed to be very similar with three-layer structured TENG, as shown in Fig. 11. We calculated the displacement (x_1 and x_2) of two masses with working frequency (ω). Assuming that the friction and gravity are excluded, and there is no damping acting on the system, we can obtain the following equations from summing forces on each mass in the horizontal direction⁵.

Figure 11 | A simple two-degree-of-freedom model consisting of two masses connected in series by two springs. Free-body diagrams of each mass in the system

$$m_1 \ddot{x}_1 + (k_1 + k_2)x_1 - k_2x_2 = 0$$

$$m_2 \ddot{x}_2 - k_2x_1 + k_2x_2 = 0$$

Using the characteristic equation for the system, the constant ω is determined as the following

$$m_1 m_2 \omega^4 - (m_1 k_2 + m_2 k_1 + m_2 k_2) \omega^2 + k_1 k_2 = 0$$

where, $m_1 = m_2 = 0.05$ kg, $k_1 = 360$ N/m, and $k_2 = 720$ N/m. For the two representative frequencies of $\omega_1 = 65$ rad/s = 10.35 Hz (the first mode) and $\omega_2 = 157$ rad/s = 25 Hz (the second mode), we can define the equation for displacement of each mass by applying Euler formulas for the sine function as follows;

$$\begin{bmatrix} x_1(t) \\ x_2(t) \end{bmatrix} = \begin{bmatrix} u_1 & u_2 \end{bmatrix} \begin{bmatrix} A_1 \sin(\omega_1 t + \phi_1) \\ A_2 \sin(\omega_2 t + \phi_2) \end{bmatrix}$$

where, the eigenvectors u_1 and u_2 are $\begin{bmatrix} 0.414 \\ 1 \end{bmatrix}$ and $\begin{bmatrix} -2.361 \\ 1 \end{bmatrix}$, and $A_1, A_2, \phi_1,$ and ϕ_2 are 0.36, -0.36, $\pi/2$, and $\pi/2$ from initial conditions of $x(0) = \begin{bmatrix} 1 \\ 0 \end{bmatrix}$ and $x'(0) = \begin{bmatrix} 0 \\ 0 \end{bmatrix}$. Finally, we can obtain the following equations.

$$x_1(t) = 0.1490 \sin\left(65t + \frac{\pi}{2}\right) + 0.851 \sin\left(157t + \frac{\pi}{2}\right)$$

$$x_2(t) = 0.35996 \sin\left(65t + \frac{\pi}{2}\right) + 0.35997 \sin\left(157t + \frac{\pi}{2}\right)$$

Fig. 12 shows the two modes of the spring-mass systems. It is clearly seen that two masses move simultaneously in same direction at low frequency of 10.35 Hz, while two masses move in opposite direction at high frequency of 25 Hz.

Figure 12 | The two modes of the spring-mass system. Displacement of two masses (a) at low frequency of 10.35 Hz and (b) at high frequency of 25 Hz.

Also, the figure 13 shows that the when two masses move simultaneously in same direction at lower frequency than 10 Hz, calculated by substituting the various frequency values to the above equation.

Figure 13 | The two modes of the spring-mass system. Displacement of two masses for low frequency at (a) 3, (b) 5, (c), 10 Hz and for high frequency at (d) 20, (e) 22, (f) 25 Hz.

(2) Comment

It is not clear how and why the author assume the areal charge density of the bottom electrode of two-layer device to be zero in the COMSOL simulation. Since the electrode is metallic, the surface should have non-zero areal charge density, such that the electric field within the metal is zero.

Author reply: We strongly appreciate reviewer's valuable and helpful comments. As shown in Fig. 14a (supplementary Fig. 2 in manuscript), when the two layers are fully separated, electrons are flowed to the bottom electrode, thus, the top electrode is positively charged and the bottom electrode is neutral. Therefore, the two layers are neutralized. This is why the areal charge density of the bottom electrode of two-layer device is assumed to be zero in the COMSOL simulation.

Figure 14 | (a) Working mechanism of two-layer structured TENG. (b, c) The electrical output current and voltage of two-layer structured TENG with 15 mm gap distance under working frequency of 3 Hz. (d, e) The electrical output current and voltage of two-layer structured TENG with gap distances from 5 mm to 15 mm under working frequency of 3 Hz.

(3) Comment

If we simply the device structure with parallel-plate model, the voltage difference between top and bottom electrode can be estimated as $\sigma \cdot d / \epsilon$. If the areal charge density of top electrode is σ on the top electrode, the voltage difference won't change with or without the third layer in the middle.

Author reply: We strongly appreciate reviewer's valuable and helpful comments. However, we are very sorry. We did not fully understand the reviewer's comment. We thought that the reviewer wanted to know the voltage difference between the top and the bottom electrode with or without the third (middle) layer. We think that the reviewer know that if there is no middle layer, the TENG is the two layer TENG. When released, the bottom electrode is neutral. With the middle layer, when released, the middle layer is neutral and positively charges are induced in bottom electrode. Thus, the potential on the bottom electrode is not zero. The maximum voltage drop in three-layer structured TENG can be defined by $\Delta V_{\text{three}} = 2\Delta V_{\text{two}} = 2\sigma d / \epsilon$, which is 2 times higher than that of two-layer TENG. (Fig. 15)

Figure 15 | The COMSOL simulations are performed for (a) the two-layer and (b) three-layer structured TENGs.

(4) Comment

The video of charging phone and watch with TENG, TENG seems stationary. How can it generate electricity and charge phone and watch without being activated? If the phone and watch are charged from a capacitor in the video, the author might want to explain explicitly in the manuscript.

Author reply: We strongly appreciate reviewer's valuable and helpful comments. In this work, the portable power-supplying system to smart watch and cellphone has two capacitors (1000 μ F) to input DC output to buck-boost circuit. The capacitors were charged by the output of three-layer structured TENGs. And then, the smart watch and cellphone were charged by the capacitors. To show it clearly, two movies, which show that the capacitors were charged by three-layer structured TENGs, followed by the batteries charging of smart phone and watch, were made and replaced in supplementary movies 6 and 7.

Movie 2, 3 | The capacitor is charged by the output of three-layer structured TENG, and the batteries of smart phone and watch are charged.

References

1. Yang, W. *et al.* 3D Stack Integrated Triboelectric Nanogenerator for Harvesting

- Vibration Energy. *Adv. Funct. Mater.* **24**, 4090–4096 (2014).
2. Lee, K. Y. *et al.* Hydrophobic Sponge Structure-Based Triboelectric Nanogenerator, *Adv. Mater.* **26**, 5037-5042 (2014).
 3. Chun, J. *et al.* Mesoporous pores impregnated with Au nanoparticles as effective dielectrics for enhancing triboelectric nanogenerator performance in harsh environments, *Energy. Environ. Sci.* **8**, 3006-3012 (2015).
 4. Lin, Z.-H. *et al.* A Self-Powered Triboelectric Nanosensor for Mercury Ion Detection. *Angew. Chem. Int. Ed.* **52**, 5065 –5069 (2013).
 5. Fay, T. H. *et al.* Coupled spring equations. *Int. J. Math. Educ. Sci.* **34**, 65-79 (2003)

Sincerely,

Jeong Min Baik, Corresponding author
School of Materials Science and Engineering,
KIST-UNIST-Ulsan Center for Convergent Materials, Ulsan National Institute of Science
and Technology (UNIST), Ulsan, 689-798, Republic of Korea
Phone: (82) 52-217- 2324
Fax: (82) 52-217- 2309
E-mail: jbaik@unist.ac.kr

REVIEWERS' COMMENTS:

Reviewer #1 (Remarks to the Author):

Authors have addressed my concerns.

Reviewer #2 (Remarks to the Author):

The revised manuscript is strengthened after the authors addressed the most of reviewers' comments. It could be considered for publication after minor revision.

In the response letter, the authors compared the output performance of two-layer TENG and three-layer TENG with and without Au particles. It seems that Au particles are a key factor for two-layer TENG, while their effect on three-layer TENG can be ignored. In other words, the proposed TENG can be further optimized by removing the Au particles. It is strongly recommended to add several sentences to claim and explain this. And Figure 4 in Response Letter should also be added to supplementary file.

Reviewer #3 (Remarks to the Author):

The author answered all my questions and addressed in the revised manuscript accordingly. This reviewer believe the manuscript is ready for publication.

August 9, 2016

Dr. Ariane Vartanian
Assistant Editor, Nature Communications
The Macmillan Building
4 Crinan Street
London N1 9XW, UK

RE: Final revisions for manuscript NCOMMS-16-10644A
Title: "Boosted Output Performance of Triboelectric Nanogenerator via Electric Double Layer Effect"
Authors: Jinsung Chun, Byeong Uk Ye, Jae Won Lee, Dukhyun Choi, Chong-Yun Kang, Sang-Woo Kim, Zhong Lin Wang, and Jeong Min Baik*
Manuscript ID: NCOMMS-16-10644A

Dear Dr. Vartanian,

We believe that the manuscript qualifies as a paper in Nature Communications. We think that we fully responded to the concerns and comments of the reviewers. Our reasons are as follows:

Reviewer #1 (Remarks to the Author):

Authors have addressed my concerns.

Author reply: We appreciate the reviewer's valuable and helpful comments.

Reviewer #2 (Remarks to the Author):

The revised manuscript is strengthened after the authors addressed the most of reviewers' comments. It could be considered for publication after minor revision.

In the response letter, the authors compared the output performance of two-layer TENG and three-layer TENG with and without Au particles. It seems that Au particles are a key factor for two-layer TENG, while their effect on three-layer TENG can be ignored. In other words, the proposed TENG can be further optimized by removing the Au particles. It is strongly recommended to add several sentences to claim and explain this. And Figure 4 in Response Letter should also be added to supplementary file.

Author reply: We appreciate the reviewer's valuable and helpful comments. We modified the output performance of two-layer TENG and three-layer TENG with and without Au particles in Supplementary Fig. 3 and added the following sentences in the 1st paragraph of the page 6.

“The Au decoration increases both output voltage and output current, although the enhancement is not significant, compared with those in the three-layer structured TENG, shown in Supplementary Fig. 3.”

Supplementary Figure 3 | Electrical outputs of triboelectric nanogenerator with and without Au nanoparticles. (a, b) The output voltages, (c, d) output currents, and (e, f) charge densities of two-layer and three-layer structured TENGs with and without Au nanoparticles.

Reviewer #3 (Remarks to the Author):

The author answered all my questions and addressed in the revised manuscript accordingly. This reviewer believe the manuscript is ready for publication.

Author reply: We appreciate the reviewer's valuable and helpful comments.

We believe that the manuscript qualifies as a paper in Nature Communications.

Sincerely,

Jeong Min Baik

Jeong Min Baik, Corresponding author

School of Materials Science and Engineering,

Ulsan National Institute of Science and Technology (UNIST), Ulsan, Republic of Korea

Phone: (82) 52-217- 2324

Fax: (82) 52-217- 2309

E-mail: jbaik@unist.ac.kr